# Photoemission of Plasmonic Gold Nanostars in Laser-Controlled Electron Current Devices for Technical and Biomedical Applications

**DOI:** 10.3390/s22114127

**Published:** 2022-05-29

**Authors:** Alexander N. Yakunin, Yury A. Avetisyan, Garif G. Akchurin, Sergey V. Zarkov, Nikolay P. Aban’shin, Vitaly A. Khanadeev, Valery V. Tuchin

**Affiliations:** 1Institute of Precision Mechanics and Control, FRC “Saratov Scientific Centre of the Russian Academy of Sciences”, 410028 Saratov, Russia; yuaavetisyan@mail.ru (Y.A.A.); akchuringg@mail.ru (G.G.A.); szarcov@gmail.com (S.V.Z.); tuchinvv@mail.ru (V.V.T.); 2Science Medical Center, Saratov State University, 410012 Saratov, Russia; 3Volga-Svet Co., Ltd., 410052 Saratov, Russia; npabanshin@mail.ru; 4Institute of Biochemistry and Physiology of Plants and Microorganisms, FRC “Saratov Scientific Centre of the Russian Academy of Sciences”, 410049 Saratov, Russia; khanadeev@gmail.com; 5Department of Microbiology, Biotechnology and Chemistry, Saratov State Agrarian University, 410012 Saratov, Russia; 6Laboratory of Laser Molecular Imaging and Machine Learning, Tomsk State University, 634050 Tomsk, Russia; 7Bach Institute of Biochemistry, FRC “Fundamentals of Biotechnology of the Russian Academy of Sciences”, 119071 Moscow, Russia

**Keywords:** gold nanostar, diamond-like carbon, surface plasmon resonance, photoinduced electron emission, hot electron, electron temperature, X-ray biosensor

## Abstract

The main goal of this work was to modify the previously developed blade-type planar structure using plasmonic gold nanostars in order to stimulate photofield emission and provide efficient laser control of the electron current. Localization and enhancement of the field at the tips of gold nanostars provided a significant increase in the tunneling electron current in the experimental sample (both electrical field and photofield emission). Irradiation at a wavelength in the vicinity of the plasmon resonance (red laser) provided a gain in the photoresponse value of up to 5 times compared to irradiation far from the resonance (green laser). The prospects for transition to regimes of structure irradiation by femtosecond laser pulses at the wavelength of surface plasmon resonance, which lead to an increase in the local optical field, are discussed. The kinetics of the energy density of photoinduced hot and thermalized electrons is estimated. The proposed laser-controlled matrix current source is promising for use in X-ray computed tomography systems.

## 1. Introduction

Plasmonic nanostructures (metallic [1,2], dielectric [3], and their composites [4,5,6,7,8]) provide strong subwavelength localizations of light. Surface plasmon resonances are supersensitive to refractive index hopping at the media interfaces. The strong local field enhancement on the sharp vertices/slits/cavities provides a mechanism for efficient resonant absorption and high photosensitivity of such structures [1,4] and their arrays [6,8]. Therefore, plasmonic nanostructures have found a wide range of applications as highly sensitive SERS-sensors [9], in electron microscopes with pico- and femtosecond temporal resolution [10], in local laser hyperthermia of biological tissues and cell transfection [11,12,13,14], and in optically controlled matrix electron emitters for 3D X-ray computer tomography (CT) [15,16,17,18].

The results of investigations carried out in the present work are connected with the development of the above-mentioned possibility of plasmonic nanostructures emitting electrons under the control of light. The last decade has been marked by the development, research, and practical application of the technology for creating arrays of field emitters based on carbon nanotubes (CNTs) for CT systems [19,20]. An array of miniature sources of electron beams based on nanostructured cathodes with field emission provided increased scanning speed and spatial resolution of CT systems. A significant increase in scan speed is achieved by reducing the switching time (low inertia) compared to systems based on hot cathodes. An increase in the number of elementary sources of electron beams and X-ray detectors (an increase in the size of the source and detector matrices) provided a significant improvement in the spatial resolution of X-ray CT systems. The entire scan is performed electronically by triggering all X-ray sources, which can be performed very rapidly. In addition to improved temporal resolution, the absence of any moving parts in stationary CT may offer improved reliability [21].

The idea of further increasing the scanning speed and spatial resolution of CT systems was developed in the implementation of optical control of the processes of generation of X-ray beams instead of electric. Recently, an X-ray source was patented [22], which includes a photocathode source of electrons. In it, the photocathode is illuminated by a spatially scanning laser light beam. An acousto-optic modulator is used to control the intensity of the laser beam and the exposure time. A photocathode based on alkali metal halides (CsBr, CsI) with low work function A ensured efficient photoemission of electrons in vacuum at photon energies hν > A.

The main goal of this work was to modify the previously developed blade-type planar structure using plasmonic gold nanostars (GNSs) [23] in order to stimulate photofield emission and provide efficient laser control of the electron current. The choice of GNSs was determined by the following: (i) high absorption of radiation compared to nanospheres [14]: (ii) lower orientation and polarization sensitivity of induced optical and temperature fields compared to nanorods [24].

## 2. Trends in the Development of Blade-Type Photosensors from Micro- to Nano-Sized Sensitive Elements

The stages of modification of the blade-type planar structure may be presented schematically as in Figure 1. Initially, the emitter was developed on the basis of a pure molybdenum blade having a submicron thickness (the design is shown in Figure 1, upper fragment). The transformation of a molybdenum blade into a composite one due to the formation of a thin nanosized diamond-like carbon (DLC) film on its surface (see Figure 1, middle fragment) was aimed primarily at stabilizing the field emission density. Such a planar structure with a composite “Mo-DLC” blade, as was theoretically justified and experimentally confirmed in [25], has a number of attractive properties, including the following:−Low voltage (providing a high field emission current density up to 100 mA/cm^2^ at a voltage of about 150 V);−Ensuring stable operation in technical vacuum conditions due to the location of the emitter and control electrodes in parallel but multi-level planes. As a result, the emitter is protected from destructive bombardment by the flow of high-energy ions of molecules of the residual atmosphere;−Emitter life of 8700 h at the above-average current density;−The technology has been worked out in the manufacture of field emission displays with a pixel matrix of 960 × 234 [26].

Both structures are characterized by high photosensitivity, which determines the prospects for their application as broadband vacuum photo-emitters [27,28]. When the emitter is irradiated with femtosecond laser pulses with a wavelength λ coinciding with the spectral position of the plasmon resonance of gold nanostars, the following two factors predominately influence the intensity of the photoemission: First, the efficiency of hot electron generation is due to local resonant absorption by certain GNS tips. Secondly, deformation in a strong optical field of the slope angle of the potential barrier profile at the metal-vacuum interface. Then the slope angle of the potential barrier profile should oscillate with the period of the incident radiation relative to the static linear slope of the barrier, which is determined by the strength of the electrostatic field in the cathode–anode gap [29].

Such structures with a metal emitter (see Figure 1, upper fragment) should potentially have a significantly faster response compared to semiconductor photodiodes. The use of composite materials such as nanofilms of diamond-like carbon (see Figure 1, middle fragment) can significantly increase the photoresponse while slightly reducing the performance. It is expected that the use of plasmonic resonance nanoparticles at the “Mo-DLC-GNS” interface, due to the plasmon focusing properties of effective local absorption of optical radiation by GNS’s tips and, accordingly, the concentration of hot electrons in a strong electrostatic field, can become effective sources of electron emission, for example, for matrix devices’ generation of local X-ray radiation and ultrafast photodetectors.

Therefore, the manufacturing technology of the first two versions of structures in Figure 1 described in [27,28] has been developed—the manufacturing technology of blade-type planar emission structures modified by gold nanostars was supplemented by the procedure of deposition of an activating layer of nanostars from a colloidal solution during the evaporation of the liquid phase.

## 3. Synthesis of Plasmonic GNS and Their Optical Properties

The synthesis of GNS was carried out according to the method described in [23]. At the first stage, according to the Frens method [30], a colloid of 19-nm gold nanospheres was prepared, which will be used further as the seeds of nanostars. In total, 1.1 mL of 1% sodium citrate was added to 50 mL of boiling 0.01% hydrochloric acid (50 mL). Within 15 min, the color of the solution changed from colorless to bright red. For the synthesis of the actual nanostars, 250 µL of 1% HAuCl_4_, 30 µL of 1 M hydrochloric acid, 500 µL of “seeds” (gold nanospheres synthesized in the previous step), 150 µL of 4 mM silver nitrate, 150 µL of 100 mM ascorbic acid. The color of the solution changed from clear to dark green within a few seconds. The nanostars were stabilized after 1 min by adding 100 µL of 1 mM thiolated polyethylene glycol and after stirring for 10 min by adding 20 µL of Tween 20.

The synthesized GNSs (see TEM image in Figure 2a) have a plasmon resonance at 800 nm and an extinction of 3.86 cm^−1^ (see Figure 2b) when suspended in water. This experimental extinction spectrum was used in the construction and tuning of the computational model. To solve the wave equation, the COMSOL software package [31] was used. At all interfaces of regions made of dissimilar materials (GNSs and surrounding water), the boundary conditions of continuity of the tangential components of the strength and the normal components of the induction of the electric and magnetic fields were satisfied. External surfaces modeling the transition to free space were constrained by PML«perfectly matched layers» [31,32].

The acceptance criterion of the computational model for describing the plasmonic properties of synthesized GNSs was the achievement of similarity between the calculated and experimental extinction spectra, both in shape and resonance wavelength. Spectral curves characterizing the evolution of the calculated spectrum of the absorption cross-section depending on the variable-length *h* of the GNS tips are shown in Figure 3a. As a result of the comparative analysis, the parameters of the computational model were determined, which are presented in Figure 3b, which meet these criteria and, in general, correspond to the results of the experiment presented in Figure 2. The calculations were performed by the finite element method at the refractive index of water *n*_m_ = 1.33, taking into account the spectral dependence of the complex refractive index of gold, borrowed from [33].

## 4. Experimental Study of Field and Photofield Emission

To conduct an experimental analysis of the effectiveness of using GNSs to stimulate the tunneling emission of an electron current source, samples of planar blade structures of the following four modifications were fabricated: with a molybdenum emitter blade (marked “Mo”, shown in Figure 4a); with a molybdenum blade and GNS coating (“Mo-GNS”, shown in Figure 4b); composite blade with DLC film (“Mo-DLC”, shown in Figure 4c); with a hybrid blade with DLC film and GNS coating (“Mo-DLC-GNS”, shown in Figure 4d).

All samples had the same electrode topology and emission structure (shown in Figure 5a). I–V measurements were carried out on an Agilent B1500A Semiconductor Device Analyzer, USA (see Figure 5b), which provides precise positioning and reliable contact of the power supply circuit elements with the switching strips of the breadboards, as shown in Figure 5c for the sample of the “Mo-DLC-GNS” structure.

Based on the results of the studies, it was found that the main difficulties in synthesizing the activating layer of GNSs in blade-type emission planar structures are associated with the probability of GNS aggregation and the creation of conductive paths that play the negative role of parasitic shunts. Depending on the degree of aggregation and the number of arising points of conjugation of the emitter and the control electrode, an ohmic contact arises, leading to the suppression of field emission, regardless of the magnitude of the applied potential difference. It turned out that most structures of the “Mo-GNS” type showed just such a defect. Despite the stability and high levels of electrical resistance of the resulting ohmic chains, the properties of the field emission after the deposition of GNS were lost, and an irreversible degradation of the emissivity occurred. In this case, the obtained I-V dependences were either linear or a function close to linear.

The main significant result of the conducted studies of the I–V characteristics was the establishment of the pattern of a significant increase in the field emission current of the structures of the “Mo-DLC-GNS” type in comparison with the emissivity of the structures of the “Mo-DLC” type. This follows from a comparison of the curves plotted in the Fowler–Nordheim coordinates in Figure 6.

An experimental study of the dependence of the tunneling emission current on the applied voltage difference U when the sensor is exposed to a low-intensity laser beam was carried out using an Agilent B1500A Semiconductor Device Analyzer, USA, shown in Figure 5b. The following CW lasers were used as radiation sources: the first one is the green laser (the second harmonic DPSS laser (Optronics, Muskogee, OK, USA) working at λ = 532 nm (photon energy, 2.33 eV) with a power *P*_532_ of 100 mW and a power density *P*_S green_ of 20 W/cm^2^. The second one is the red laser (OGLD 03, China) working at λ = 662 nm (photon energy, 1.87 eV) with a power *P*_662_ of 20 mW and a power density *P*_S red_ of 1 W/cm^2^. The results are presented in Figure 7. From the analysis of the calculated spectral absorption curve of GNSs in the air (taking into account the negligible differences in the refractive indices in vacuum and air, we assume that a similar curve for vacuum will be valid) in Figure 7, it follows that the wavelength of the red laser λ = 662 nm almost coincides with the maximum of the plasmon resonance of GNS used in the experiment. As the green laser wavelength λ = 532 nm corresponds to absorption far from resonance, where GNS absorption cross-section is 11 times lower. Under such conditions, simple estimates “from above” give a value of no more than 24 K for the increase in temperature in the case of irradiation with a green laser and no more than 12 K with a red one.

As can be seen from the data in Figure 7, the relative increment of the photocurrent (determined by the difference between the tunnel current *I*_laser_ of the sensor-emitter under laser irradiation and the tunnel current *I*_dark_ in the absence of irradiation) increases monotonically with increasing *U*. The increment of the photocurrent (*I*_laser_ − *I*_dark_) is normalized to the value of the maximum *I*_0_ of the dark current, registered when changing *U* in the studied range. It follows from Figure 7 that even at relatively low *U*, the photocurrent reaches almost 18% of the field emission current *I*_0_ when irradiated with a green laser and 21% when irradiated with a red laser. In both cases, there is a tendency for the photocurrent to further increase with increasing *U*. Despite the fact that the exposure power *P*_532_ of the green laser is 5 times higher than the power *P*_662_ of the red one, the photocurrent in the latter case turned out to be higher. This is clear experimental proof of the efficiency of using plasmonic nanoparticles irradiated by a laser beam at a resonant wavelength to generate tunneling photoelectrons. It should be especially noted that this result was obtained in the regime of irradiation with a low-power CW laser source, and the photon energy is much less than the work function of electrons from the metal.

For the convenience of comparative evaluation of the proposed “Mo-DLC-GNS” structure with analogues in terms of a set of parameters that determine their suitability for use in high-performance and safe X-ray CT systems, the information is summarized in Table 1.

It is known that the main limitation of generating a significant photoinduced current is the unacceptably high heating of the emitter as the intensity of the laser beam increases. Therefore, the use of pulsed laser radiation in the mode of short pulses instead of CW can become a universal recipe for realizing the possibility of deep optical modulation of the laser-induced electron current. Figure 8 schematically illustrates the periodic change in the profile of the potential barrier under the simultaneous influence of strong electrostatic (intensity F_0_) and optical (amplitude F_L_) fields. The change in the width and height of the potential barrier with the frequency of optical vibrations ω has a significant effect on the permeability of the barrier, determining the magnitude of field and photoemission.

As successful examples of this approach, we can cite the results of [28,34,35,36,37]. Thus, it was experimentally demonstrated in [29] that upon activation of a single needle by a laser beam with a mean power density *P*_S_ of 8 GW/cm^2^ at a wavelength of 800 nm, a single 8 fs-pulse is capable of producing a single electron pulse of less than 1 fs-duration. Therefore, the main application of such a structure is associated with high-resolution, precision electron-beam spectroscopy both in time and space.

As described in [35], electron pulses from a tungsten nanotip with a duration of up to 250 fs and a repetition rate of 1 MHz were obtained by controlling them with laser pulses of 350 fs and a wavelength of 515 nm. The choice of the parameters of the pulsed mode was carried out by the authors based on the possibility to provide stable (without significant degradation) operation of an ultrafast high-emittance electron gun for several hours.

Previously, it was experimentally shown [28] that up to the power density of *P*_S_ = 1 MW/cm^2^, single-photon laser-induced emission (linear dependence of the photocurrent on the power *P* or power density *P*_S_) of the blade type “Mo-DLC” structure is observed. Therefore, the result obtained in this study (see Figure 6 and Figure 7, *P*_S_ was in the range of 1–20 W/cm^2^) makes it possible to predict the prospect of a significant increase in the photosensitivity of the “Mo-DLC-GNS” structure with a developed surface in proportion to the increase in *P* or *P*_S_ of the laser. Experimentally, the possibility of multiple amplification of the photoemission current by connecting various channels of multiphoton photoexcitation at wavelengths from infrared to ultraviolet was studied in [36,37]. The photoemission sources were composite structures—nanodiamond-coated tungsten tips [34] and single-crystal diamond needles [35]. However, these effects of multiphoton excitation appear and become dominant at very high laser radiation intensities, amounting to tens of GW/cm^2^ or more. In the next subsection, the principal features of the application of femtosecond pulsed laser irradiation are considered.

## 5. Estimation of Admissible Regimes of GNS Irradiation with Femtosecond Pulses

Thermal processes accompanying the interaction of intense optical radiation with the emitter material, along with the stimulation of optical field emission, can significantly limit the performance of such a current source due to melting, ablation, and Coulomb destruction of the emitter material [24,38,39]. Therefore, for a reasonable assessment of the regimes acceptable for the practical application of plasmonic nanoparticles in systems with photoemission, an integrated model for the joint calculation of the processes of interaction of laser radiation with nanoparticles is proposed. The model of the formation of the optical field and the distribution of the absorbed radiation power, constructed in [15], should be supplemented by a system of equations describing the kinetics of changes in the temperature of elementary emitters.

To simulate the transformation of the energy of irradiating femtosecond laser pulses into thermal energy in the GNS, we used the approach proposed in a recent paper [40]. Note that under impulsive photoexcitation of bulk materials and nanoparticles, a non-thermal electron population is generated. Such an out-of-equilibrium gas (so-called hot electrons) is characterized by a broad energy distribution that thermalizes on a sub-picosecond time scale via electron–electron and electron–phonon collisions, giving rise to a high-temperature electron gas. In contrast to the so-called two-temperature models (see, for example, refs. [41,42,43] and the literature cited there), where the kinetics of the temperature of thermal electrons and the lattice is analyzed by taking into account the electron-phonon interaction, in the work [40] additionally takes into account explicitly electron-electron collisions, which lead to energy relaxation of the fraction of photoexcited hot electrons. The corresponding system of equations can be reduced to the following form:(1)$$\gamma \vartheta_{e}\frac{d\vartheta_{e}}{dt}=P_{a}-G(\vartheta_{e}-\vartheta_{l})-\frac{dN}{dt}$$
(2)$$C_{l}\frac{d\vartheta_{l}}{dt}=G(\vartheta_{e}-\vartheta_{l})$$

Here *t* is the time, $\vartheta_{e}$ is the temperature of thermal electrons, $\vartheta_{l}$ is the temperature of the atomic lattice of the nanoparticle material, *N* is the energy density of photoexcited electrons, *P*_a_ is the specific absorbed power of laser radiation; *γ* is the heat capacity constant of electrons; *G* is the electron-phonon interaction constant; *C*_l_ is the volumetric heat capacity of the lattice. For a rectangular excitation pulse of duration *t*_p_, based on the equations of [36], the kinetics *N*(*t*) can be written explicitly as follows:(3)$$N(t)=\left\{ \begin{array}{l} \tau_{ee}P_{a}[1-\exp(\frac{-t}{\tau_{ee}})]\;\mathrm{npu}\;t\leq t_{p},\\ \tau_{ee}P_{a}\exp(\frac{-t}{\tau_{ee}})[\exp(\frac{t_{p}}{\tau_{ee}})-1]\;\mathrm{npu}\;t\geq t_{p}, \end{array}\right.$$
where *τ_ee_* is the electron-electron relaxation time constant. The transition to the two-temperature model is obtained by neglecting the last term in Equation (1), i.e., by the substitution *dN*/*dt* = 0.

The limitations of the model (1)–(3) are, firstly, the absence of spatial dependence of all dynamic variables, which are interpreted as average values over the volume of the analyzed nanoparticle; secondly, the heat exchange between the nanoparticle and the environment is neglected (which can be easily verified by summing Equations (1) and (2)), which corresponds to adiabatic boundary conditions. Finally, we note that the approach we used [36] describes the effect of the hot electron density N(t) only on the kinetics of the temperature $\vartheta_{e}(t)$ of thermalized electrons and the lattice temperature $\vartheta_{l}(t)$, neglecting the effect N(t) on the photocurrent.

At the same time, the spatial distribution of thermal sources photoinduced in the GNS is essentially spatially inhomogeneous, being localized mainly in the tips [15], where pronounced regions of absorption localization are also observed. The latter is clearly demonstrated in Figure 9. The finite element model described in [15] was used, the dimensional parameters of the GNS were determined in Section 2, the calculation for the complete system considering the air environment was carried out in the COMSOL environment [31], and the distribution topograms are given for the tip with maximum absorption.

Figure 9a clearly shows that the maximum specific volumetric absorption power values of *P*_a_ are not at the top (end) of the tip but are shifted towards the base. They are located on the surface of the tip, forming a conical annular segment and thus determining the region of localization of radiation absorption.

Further, along with a significant spatial inhomogeneity of the absorption of the optical field in the GNS tips, Figure 9b shows a noticeable inhomogeneity in the distribution of the field component *E*_n_, normal to the surface of the tip, which determines the emission of hot electrons [44,45,46]. According to these works, the larger the value of *E*_n_, the higher the intensity of the emission of hot electrons. From Figure 9b, it follows that the normal field component is indeed maximal near the tip end, but the concentration of hot electrons in this region is low. The noticeable value of the normal component of the field in the region of the maximum concentration of hot electrons (the aforementioned large-area cone annular segment) allows us to make an assumption about a significant additional contribution of this region to the emission current; thereby, increasing the photosensitivity of the sensor/emitter.

To partially overcome the first limitation, which is a pronounced spatial inhomogeneity of the distribution of heat sources *P*_a_ in the tips of a nanostar, we proposed the following. Select a pronounced area of *P*_a_ localization and, neglecting the heat exchange of this area with the environment, consider Equations (1) and (2) acceptable for dynamic variables, now considered as average values over the volume of this area of *P*_a_ localization.

To estimate the error introduced into the solution of the equations by the adiabatic condition, we additionally solved these equations under the assumption of the maximum possible thermal contact of the nanoparticle lattice with the environment. Namely, it was assumed that $\vartheta_{l}$(*t*) = const over the entire considered time interval, which corresponds to the isothermal boundary condition. It seems that the solution of equations with a more adequate real boundary condition will be in the range between the solutions corresponding to these two boundary conditions. The results of the numerical solution of Equations (1) and (2) are shown in Figure 10, the data given in [39,40] on the temperature of the Coulomb explosion and the parameters of the GNS material were used.

According to the data in Figure 10, three stages can be distinguished in the electron temperature $\vartheta_{e}$ kinetics as follows: one during heating and two during cooling of the electron fraction. The maximum of the electron temperature $\vartheta_{e}{}^{\max}$ is reached at the moment of time $t_{e}^{\max}$, which is much greater (approximately twenty times) than the pulse duration *t*_p_. Then, at the second stage of cooling, a very long decrease in $\vartheta_{e}$ is observed according to an almost linear law. Additionally, finally, at the third and final stage of cooling, close to the curves reaching a plateau (we denote the estimate for this time as $t_{e}^{fin}$), the value $\vartheta_{e}$ decreases according to an exponentially similar law. Under the adiabatic boundary condition, the lattice temperature $\vartheta_{l}$ increases monotonically and, together with $\vartheta_{e}$, reaches the temperature level corresponding to the absorbed energy density *P*_a_·*t*_p_.

Note that, according to Figure 10, the solutions corresponding to the adiabatic and isothermal boundary conditions are very close over a significant part of the time interval, excluding only the final stage of cooling of the electron fraction. According to the above reasoning about the “fork” between the solutions corresponding to these two alternative boundary conditions, the adopted calculation method can be considered acceptable for estimating the electron and lattice temperature kinetics.

According to the results in Figure 10, a comparative analysis of the kinetics of the density N(t) of hot electrons and the density $N_{e}(t)=0.5\gamma {\left[\vartheta_{e}(t)\right]}^{2}$ of thermalized electrons leads to the following. At the initial time N(t=0)=0, while $N_{e}(t=0)=3.06\times 10^{6}$ J/m^3^. In the course of time, N(t) obtains advanced development and at the moment *t* = *t*_p_ = 100 fs, the value $N(t_{p})=1.6\times 10^{9}$ J/m^3^ turns out to be maximum and noticeably exceeds $N_{e}(t_{p})=2.04\times 10^{8}$ J/m^3^. At the moment of time $t_{e}^{\max}=1.8$ ps corresponding to the maximum energy density of thermalized electrons $N_{e}(t_{e}^{\max})=1.7\times 10^{9}$ J/m^3^, the energy density of hot electrons turns out to be significantly lower as follows: $N(t_{e}^{\max})=2.5\times 10^{7}$ J/m^3^. Thus, the ratio $N(t)/N_{e}(t)$, which is small at the initial stage of irradiation, for two points in time, *t*_p_ = 100 fs, and $t_{e}^{\max}=1.8$ ps is approximately 8 and 0.015, respectively.

Based on an approximate analytical analysis of the solutions of Equations (1) and (2) for the specified characteristics $t_{e}^{\max}$, $\vartheta_{e}{}^{\max}$, $t_{e}^{fin}$, we have obtained the following formulas:(4)$$t_{e}^{\max}\approx \tau_{ee}\ln\left\{\frac{\vartheta_{0}+\frac{P_{a}}{G}[\exp(\frac{t_{p}}{\tau_{ee}})-1]}{\vartheta_{e}^{\max}}\right\},$$
(5)$$\vartheta_{e}^{\max}\approx \sqrt{\vartheta_{0}^{2}+\frac{2P_{a}t_{p}}{\gamma }},$$
(6)$$t_{e}^{fin}\approx t_{e}^{\max}+\frac{\gamma }{G}(\vartheta_{e}^{\max}-\vartheta_{0}),$$
where $\vartheta_{0}=\vartheta_{e}(t=0)=\vartheta_{l}(t=0)$, giving acceptable estimates of the corresponding characteristics. So, it can be seen from Figure 11 that as *P*_S_ increases with a constant step of 50 MW/cm^2^, the values $\vartheta_{e}{}^{\max}$ increase not in proportion to *P*_S_ but approximately as *P*^1/2^, which agrees with Formula (5).

Figure 12 shows the kinetics of the electron temperature at a constant value of the product *P*_S_*t*_p_, and hence *P*_a_*t*_p_, that is, at the same value of the energy density absorbed by the tip, but at different values of *P*_S_ and *t*_p_. It can be seen that the advanced development, which, naturally, the kinetics receives at a larger value of *P*, quickly levels out and almost completely disappears at the time $t_{e}^{\max}$ of maximum $\vartheta_{e}$.

Thus, the maximum electron temperature of GNS is determined by the pulse energy of the irradiating laser beam (or the energy absorbed during the pulse). This expands the possibilities of robust control of the irradiation regime due to the synchronous variation of two parameters—the laser beam power density and the pulse duration.

Figure 13 compares the results of the following calculations: in model 1, a two-temperature model that neglects the kinetics of the hot electron density *N*(*t*); (substitution in Equation (1) *dN*/*dt* = 0); in model 2, based directly on Equation (1), which explicitly takes into account the kinetics of *N*(*t*).

Analysis of the results of Figure 13 allows us to draw the following conclusions:The time to reach the maximum electron temperature $t_{e}^{\max}$ in model 2 is significantly (approximately 20 times) higher than this value for the two-temperature model;At the same power density of irradiating light, the maximum of electron temperature $\vartheta_{e}{}^{\max}$ in model 2 is approximately 8% less than this value for the two-temperature model;At the same value of the electron temperature maxima $\vartheta_{e}{}^{\max}$, the time interval during which the electron temperature retains high values is approximately three times higher than this value for the two-temperature model as follows: $\Delta t_{2}/\Delta t_{1}\approx 3$ (see inset in Figure 13).

Thus, the analysis of the dynamics of energy exchange processes using the described model and the nature of possible mechanisms of destruction [24,38,39] of the elements of the emitter structure with optical stimulation of emission leads to the formulation of a system of criteria. Their implementation will ensure the successful solution of the controversial problem—strictly dosed (precision) and ultra-fast control of the level of the electron flow while maintaining the necessary parameters of the device’s durability.

## 6. Conclusions

The physical and technological aspects of the use of GNSs for activating the tunnel emission of an electron current source based on planar-end blade structures [15] with a developed surface are considered in the context of possible applications in the field of biomedical instrumentation and sensing, including the design of a matrix source with photoemission for X-ray computed tomography. It is shown that among the experimentally studied variants of emission structures with GNS, the variant of the structure “Mo-DLC-GNS” with a nanosized DLC film preliminarily formed on the emitter blade has the highest manufacturability and operational stability.

The results of an experimental study of the field emission properties of structures with and without deposited GNS confirmed the conclusions of the previous theoretical analysis [15] on the enhancement of the electrostatic field up to six times and the stimulation of field emission as follows: it has now been demonstrated that the localization and enhancement of the electrostatic field at the ends of GNS’s tips increases the tunneling current of the structure up to two orders of magnitude in the CW mode at the same applied voltage.

A mathematical model has been constructed for calculating the optical properties of plasmonic GNS, the adequacy of which is ensured by the correspondence of the simulation results to the experimental data on the measurement of the extinction spectrum in water. The wavelength of the surface plasmon resonance GNS in the air is determined to be λ = 665 nm.

Significant and comparable in magnitude, photoemission current was recorded upon irradiation with a low-power laser both at a wavelength in the vicinity of the plasmon resonance (red laser) and far from the resonance (green laser). Despite the fact that the exposure power density of the green laser was 5 times higher than the exposure power of the red one, the photocurrent in the latter case turned out to be higher (18% and 21% of the field emission current *I*_0_, respectively, for the red and green lasers). This is a clear experimental proof of the efficiency of using plasmonic nanoparticles irradiated by a laser beam at a resonant wavelength to generate tunneling photoelectrons.

The prospects and conditions for the transition to modes of irradiation of the structure with femtosecond laser pulses at the surface plasmon resonance wavelength to increase the photocurrent by several orders of magnitude and ensure deep optical modulation of the electron current while maintaining the operability of emission devices are discussed. An integrated model for the combined analysis of optical processes and electron temperature kinetics is proposed. On this basis, an estimate of the admissible modes of femtosecond pulsed irradiation of structures with gold plasmonic nanostars was performed. It is shown that taking into account electron-electron interactions significantly performed the kinetics of the electron temperature and, accordingly, affects the parameters of the maximum allowable irradiation regimes. The kinetics of the energy density of photoinduced hot and thermalized electrons is estimated.

## Figures and Tables

**Figure 1 sensors-22-04127-f001:**
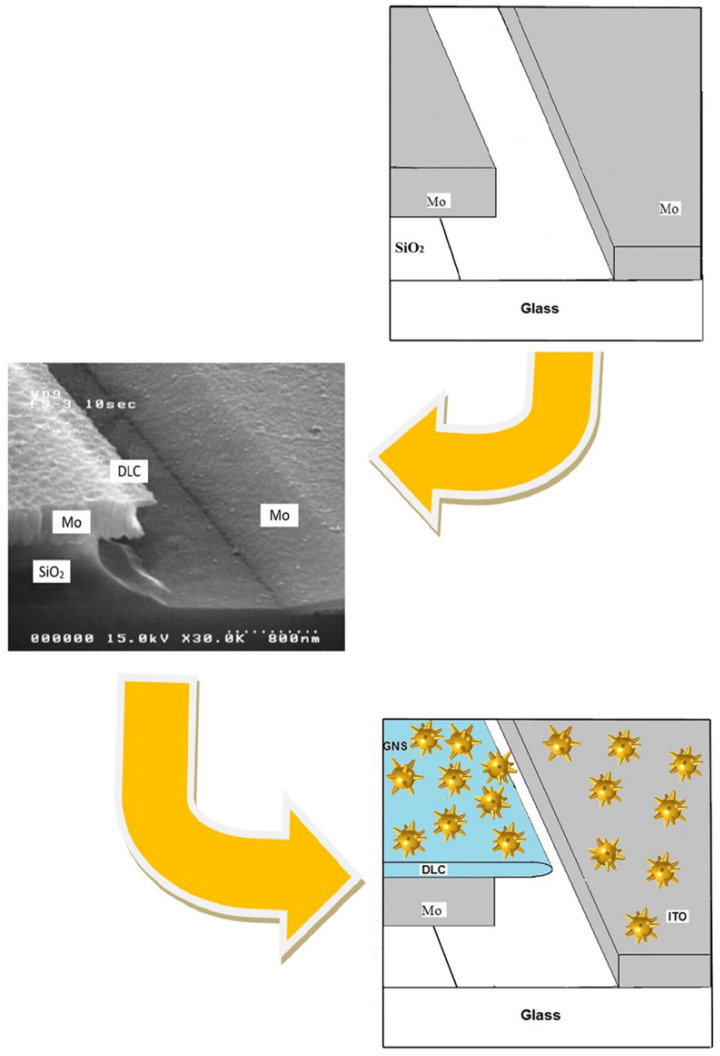
Stages of blade-type emission structure modification: with molybdenum (**upper** fragment), composite “Mo-DLC” (**middle** fragment) and hybrid “Mo-DLC-GNS” (**lower** fragment) blades.

**Figure 2 sensors-22-04127-f002:**
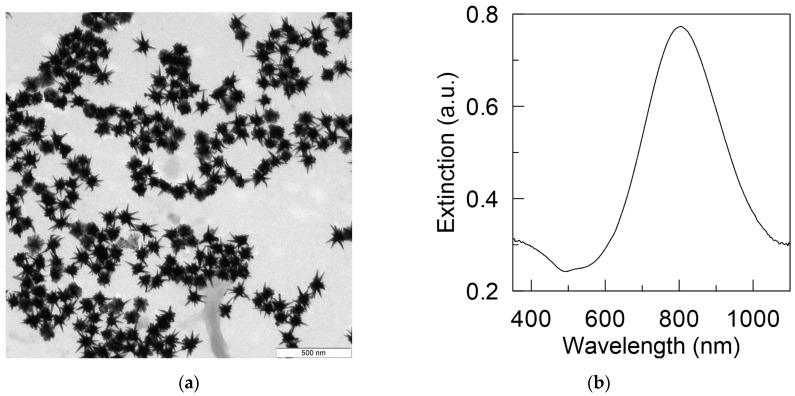
TEM image (**a**) and extinction spectrum (**b**) of synthesized GNS suspended in water.

**Figure 3 sensors-22-04127-f003:**
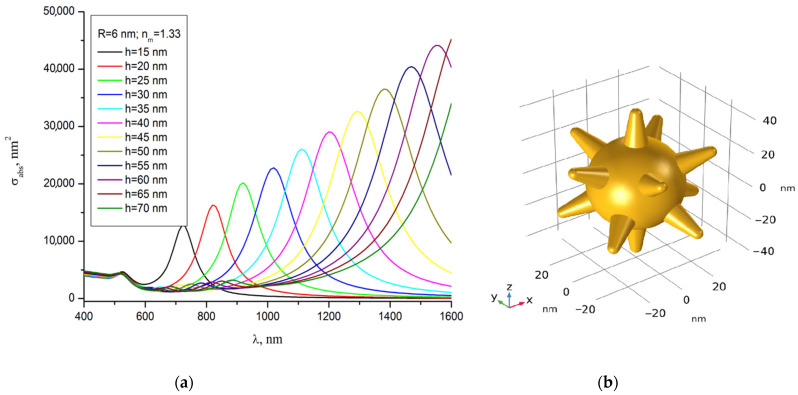
Calculated absorption cross-section spectra depending on the length *h* of the tips of GNS in water: the GNS is irradiated with laser light linearly polarized along the axis of one of the tips (**a**). Determined geometric model of the GNS: the core diameter is 50 nm, the length *h* of the conical tips is 19 nm, the diameter of the base of the tips is 12 nm, the radius of the tip end is 3 nm, the number of tips is 12 (**b**).

**Figure 4 sensors-22-04127-f004:**
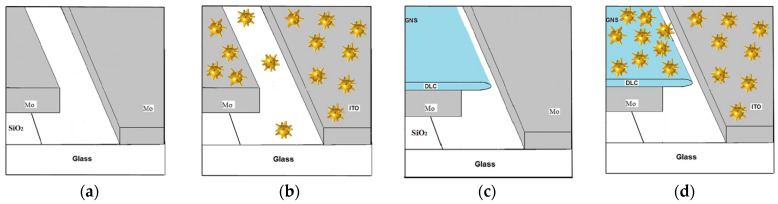
Structure of experimental samples: “Mo” (**a**); “Mo-GNS” (**b**); “Mo-DLC” (**c**); “Mo-DLC-GNS” (**d**).

**Figure 5 sensors-22-04127-f005:**
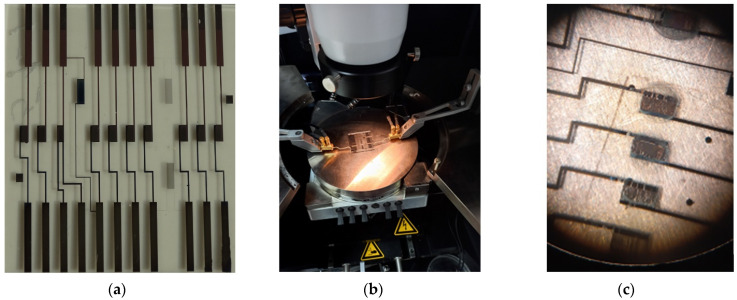
Experimental arrangement for study of field and photofield emission: photo of the layout with emission structures “Mo-DLC” (**a**); experimental setup for measuring the I–V characteristics with a mock-up structure “Mo-DLC-GNS” (**b**); photo of the emission zone of the layouts of the “Mo-DLC-GNS” structure, obtained using an optical microscope of the probe station (**c**).

**Figure 6 sensors-22-04127-f006:**
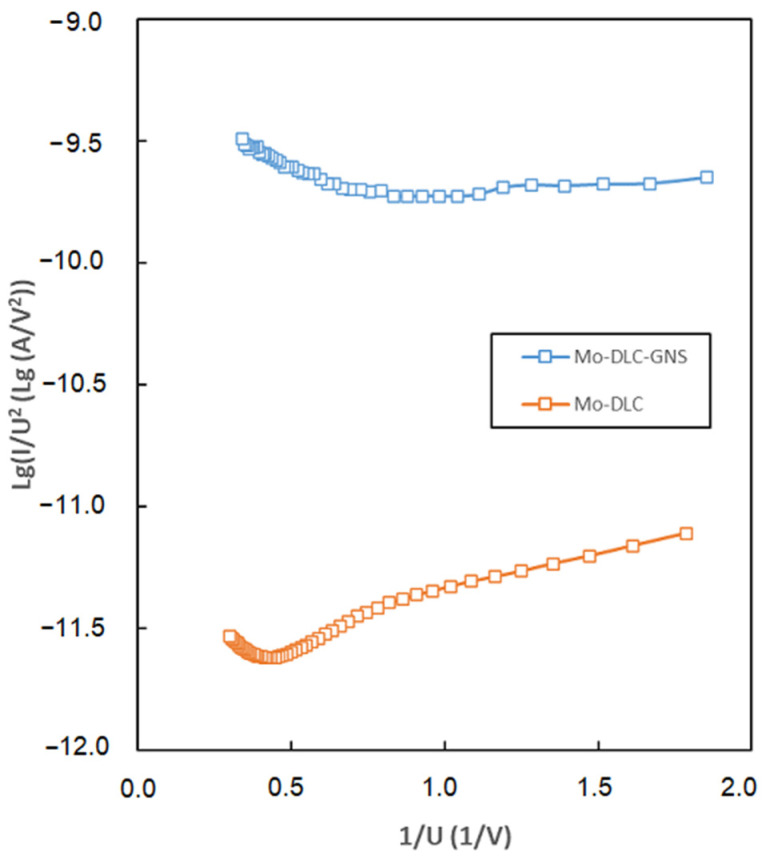
Results of measurement of volt-ampere characteristics plotted in Fowler–Nordheim coordinates for emission structures of two types—“Mo-DLC” and “Mo-DLC-GNS”.

**Figure 7 sensors-22-04127-f007:**
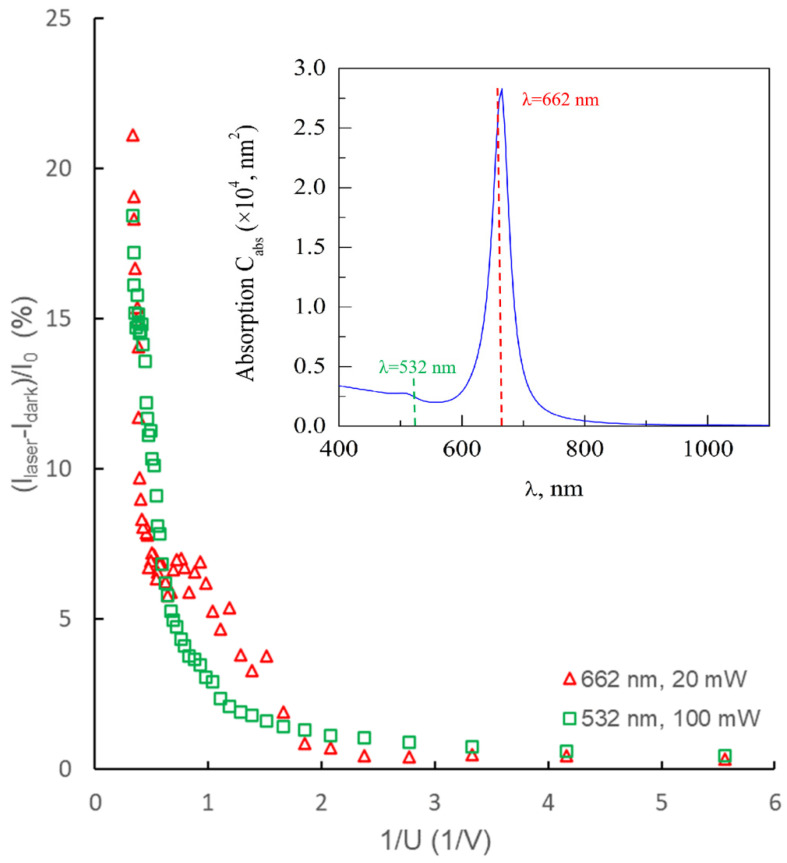
Dependence of the photocurrent, normalized to the value of the maximum dark current *I*_0_, with a change in the value 1/*U*: square markers correspond to red laser irradiation (λ = 662 nm, laser power *P*_662_ of 20 mW, triangular markers correspond to green laser irradiation (λ = 532 nm, laser power *P*_532_ of 100 mW. The inset shows the absorption spectrum of GNPs in air, where the wavelengths of the used lasers are shown.

**Figure 8 sensors-22-04127-f008:**
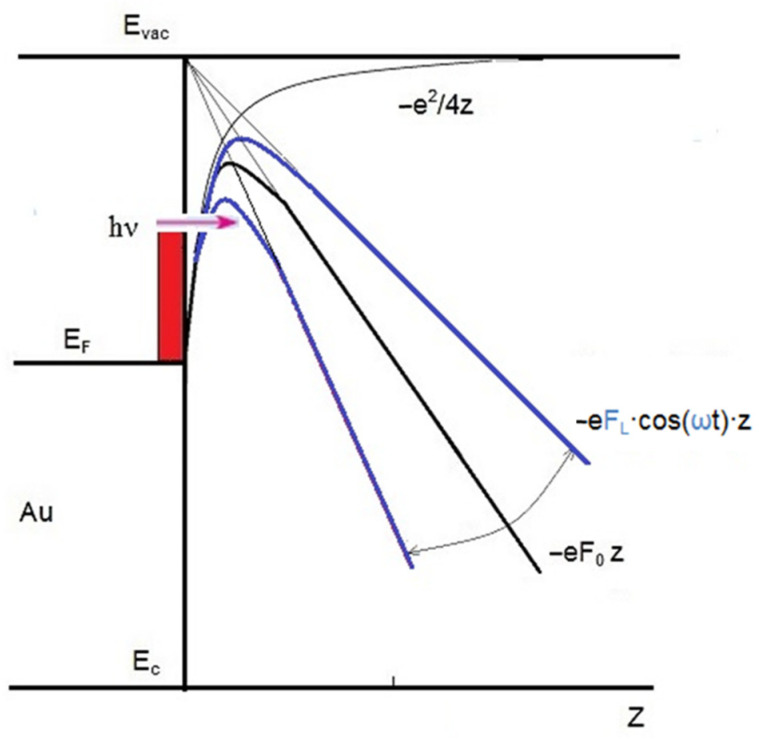
Scheme of energy levels near the GNS surface and the form of a potential barrier in a strong electrostatic field F_0_ and an oscillating laser field F_L_ with allowance for the Schottky effect for field and photoemission of hot electrons with energy hν.

**Figure 9 sensors-22-04127-f009:**
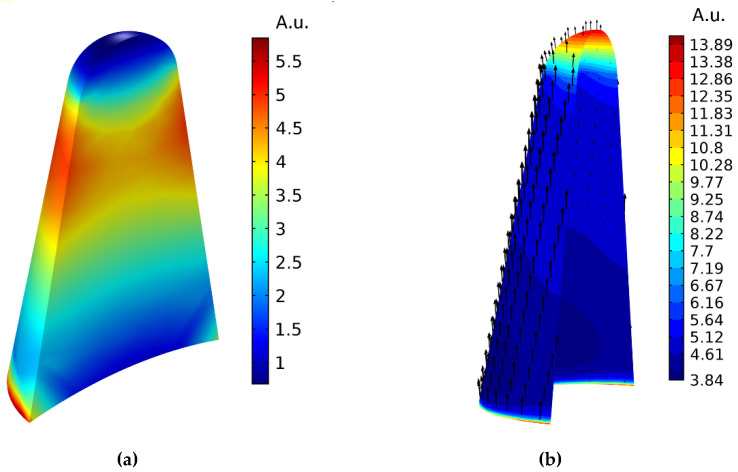
Regions of absorption localization: topogram of the intensity distribution of the radiation absorption density in the central section of the GNS tip shown in Figure 3b (**a**); a schematic representation of the distribution of the electric field vector on the inner surface of the tip (inside GNS) (**b**). GNS is irradiated with laser light linearly polarized along the tip axis at a plasmon resonance wavelength of 665 nm.

**Figure 10 sensors-22-04127-f010:**
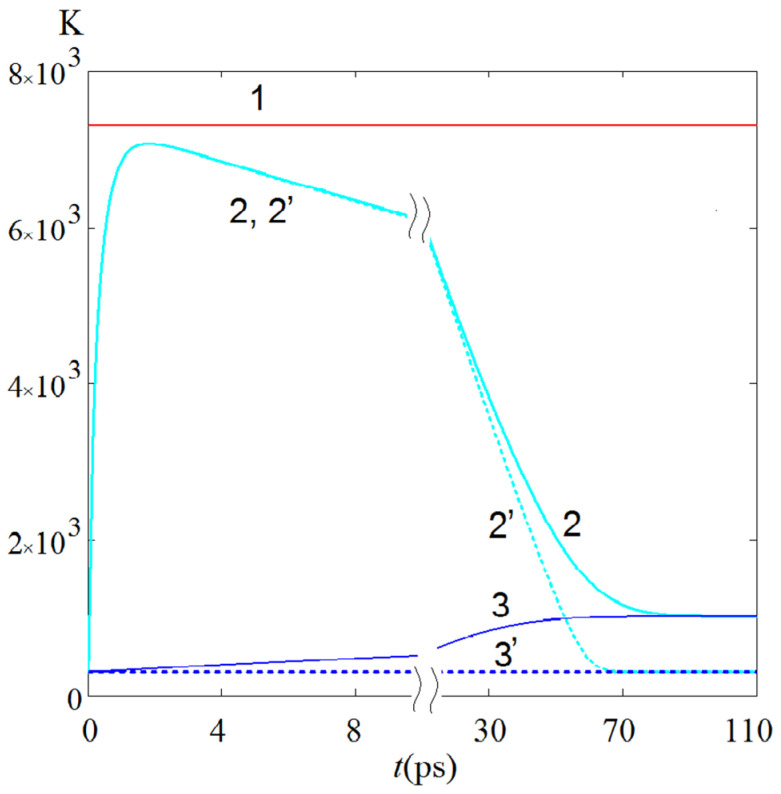
Calculated kinetics of the electron temperature $\vartheta_{e}$ (curves 2, 2′) and crystal lattice temperature $\vartheta_{l}$ (curves 3, 3′) in the region of radiation absorption localization in the tip of the considered GNS under the influence of a rectangular laser pulse of duration *t*_p_ = 100 fs, power density *P*_S_ = 250 MW/cm^2^ at a resonant wavelength of 665 nm. The solid (dashed) curves correspond to the adiabatic (isothermal) boundary condition at the initial condition *N*(*t* = 0) = 0, $\vartheta_{e}(t=0)=\vartheta_{l}(t=0)=300\;K$; the solid red level (marked with the number 1) is the temperature of the Coulomb explosion. The results were obtained with parameters: *τ_ee_* = 410 fs, *γ* = 68 J m^−3^ K^−2^, *G* = 0.91 × 10^16^ W m^−3^ K^−1^, *C*_l_ = 2.5 × 10^6^ J m^−3^ K^−1^.

**Figure 11 sensors-22-04127-f011:**
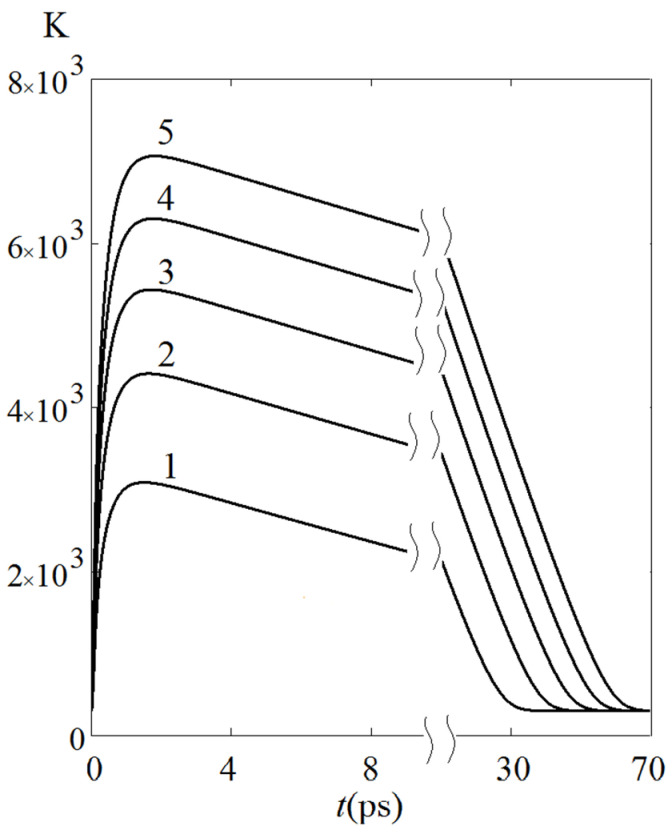
Kinetics of electron temperature when varying the power density of the irradiating pulse in the range *P*_S_ = 50 MW/cm^2^ (curve 1) − 250 MW/cm^2^ (curve 5), fixed laser pulse duration *t*_p_ = 100 fs, isothermal boundary condition.

**Figure 12 sensors-22-04127-f012:**
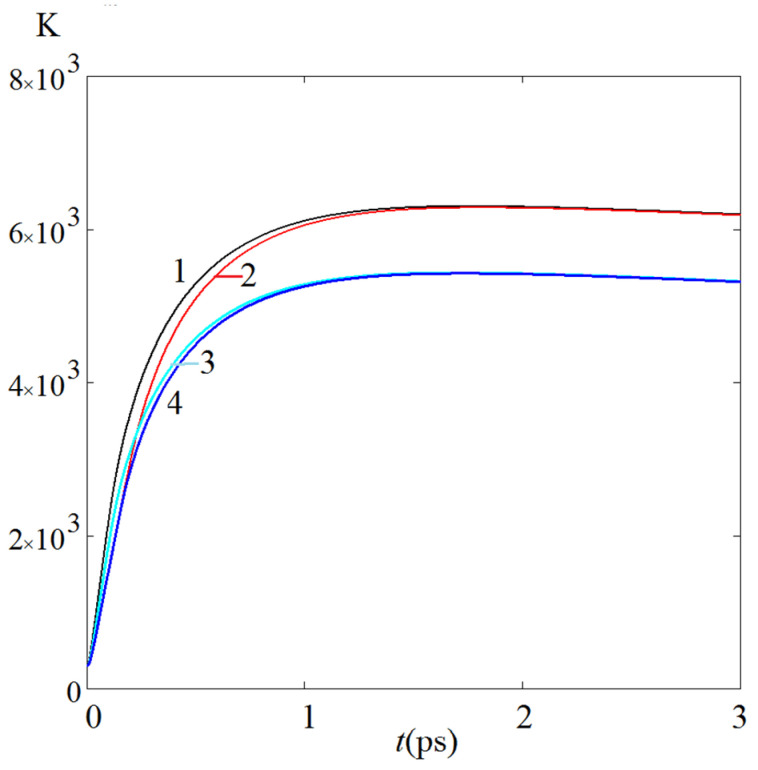
Kinetics of electron temperature for the following parameters: curve 1—power density of irradiating light *P*_S_ = 200 MW/cm^2^ and pulse duration *t*_p_ = 100 fs; curve 2—100 MW/cm^2^ and 200 fs; curve 3—150 MW/cm^2^ and 100 fs; curve 4—100 MW/cm^2^ and 150 fs, respectively. The calculations correspond to the isothermal boundary condition with the same other parameters.

**Figure 13 sensors-22-04127-f013:**
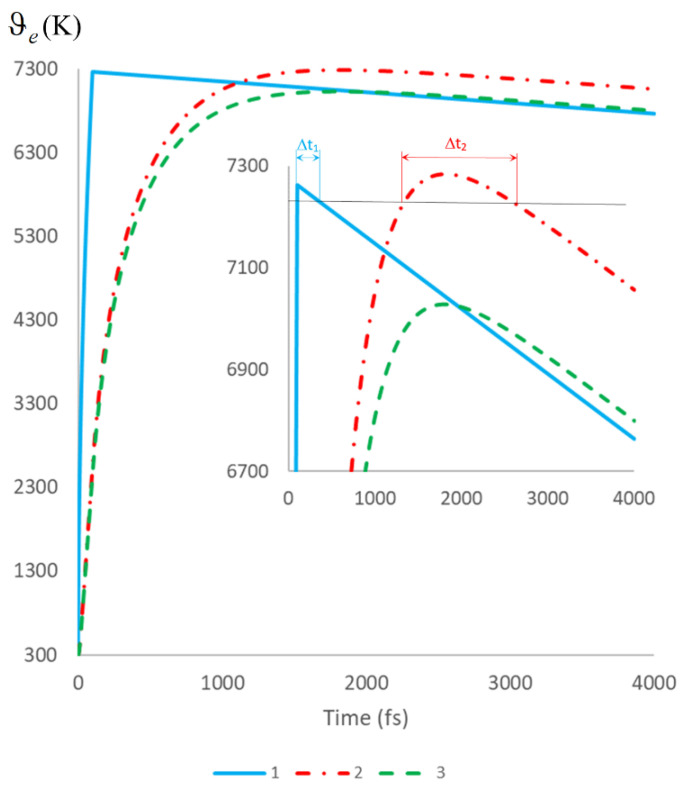
Electron temperature $\vartheta_{e}$ kinetics, calculated for the duration of the irradiating pulse *t*_p_ = 100 fs and the following parameters: curve 1—power density of the irradiating light *P*_S_ = 250 MW/cm^2^, two-temperature model; curve 2—*P*_S_ = 270 MW/cm^2^, model 2; curve 3—*P*_S_ = 250 MW/cm^2^, model 2. The calculations correspond to the isothermal boundary condition with the same other parameters.

**Table 1 sensors-22-04127-t001:** Comparison of the parameters of non-incandescent matrix sources of electron current for X-ray CT.

Parameter	Ref. [19]	Ref. [22]	Proposed in This Paper
Characteristics of the emission structure	Array of vertically aligned CNT bundles	Metal film doped with CsBr, CsI	Array of blade-type planar pixels “Mo-DLC-GNS”
Period of elementary emitter arrangement	15 µm	-	6 µm
Pixel size	2.0 mm × 0.5 mm	1.0 mm × 1.0 mm	1.0 mm × 0.3 mm
Number of pixels	-	Hundreds	up to 960 × 234 pixels
Possibility of optical control	-	+	+
Operating voltage required to provide photofield emission	2.1 kV electrical field emission	250 kV photofield emission	≤150 V photofield emission
Wavelength of control laser radiation	-	405 nm	532 nm (green)/ 665 nm (red)
Average power of the laser controlling the emitter current	-	150 mW	100 mW (green)/ 20 mW (red)

## Data Availability

Data is contained within the article.

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
