# Peer review of "Photoemission of Plasmonic Gold Nanostars in Laser-Controlled Electron Current Devices for Technical and Biomedical Applications"

_sensors, 2022, doi:10.3390/s22114127_

Round 1
Reviewer 1 Report
In this paper, the authors discuss the use of gold nano stars to activate electron current source tunneling emission. It has been found that the deposition of a film of diamond-like carbon on the emitter blade contributes to the stabilization of the functioning of structures using gold nanostars. This paper is well written, but needs the following revisions before publication:
- What are the advantages of this job over other jobs? The author is advised to make a table for comparison.
- The biggest problem of this paper is that the format of the article is messy, which needs to be greatly adjusted according to the way of scientific papers.
- In the calculation part of Figure 3, the author needs to give detailed calculation parameters, simulation environment, and calculation methods.
- About “surface plasmon resonance”, some relevant literature authors need to mention, such as: RSC Adv., 2022, 12, 7821-7829; Appl. Phys. Express 12, 052015 (2019); Plasmonics 2014, 9, 1163–1169; Physical Chemistry Chemical Physics, 2022, 24, 4871 – 4880; Plasmonics 2018, 13, 345–352.
Reviewer 2 Report
The paper by Yakunin, Avetisyan, Akchurin, Zarkov, Abanshin, Khanadeev, and Tuchin is devoted to the influence of gold nano stars (GNS) on activating of the tunnel emission of the electrons in planar-end blade structures. It is experimentally shown that GNS, deposited on diamond like carbon blade, increases the tunneling current of the structure up to two orders of magnitude in the continue current mode. Then an enhancement of the photoemission is investigated in the presence of GNS. Significant and comparable in magnitude photoemission current was recorded upon irradiation with a low-power laser both at a wavelength near the plasmon resonance at a GNS (red laser) and far from the resonance (green laser). Yet, exposure power density of the green laser was 5 times higher than the exposure power of the red laser. Authors consider this observation as a clear experimental proof of the efficiency of using plasmonic nanoparticles irradiated by a laser beam at a resonant wavelength in generating tunneling photoelectrons. It is an interesting experimental work that could be published.
I have a comment on theoretical part of the paper, where a model for the energy balance in GNS is used to calculate the evolution the electron temperature under the influence of the laser radiation. It looks like the authors assume two sorts of electrons, namely, there are thermalized electron and so called photoexcited electrons. It will be useful to explicitly state existence of hot electrons and discussed its relative density and its influence on the photocurrent. Does the photocurrent has sharp maximum during the time when the thermalized electrons are excited?
Reviewer 3 Report
Comments on the manuscript
In this manuscript entitled “Photoemission of Plasmonic Gold Nanostars in Laser Controlled Electron Current Devices for Technical and Biomedical Applications” the authors presented theoretical and experimental results on the enhanced electron current enabled by the sharp tips of plasmonic nanostars. They also included systematic studies on the photofield emission and irradiation of the nanostars with ultra-fast dynamics using femtosecond pulses. The idea of this manuscript is, in general, interesting and the theoretical results are incremental to the field of nano-plasmonics. The manuscript is technically sound with well-supported conclusions and assertions. However, the language is a little difficult to understand with some flaws in fluency and accuracy, which degrades the whole level of the manuscript.
Thus, this manuscript needs substantial revision before it can meet the scope of Sensors. Consequently, there are a few major/minor points that should be addressed for the purpose of clarity.
- It is very difficult to understand the motivation for this work. The abstract is very disorganized, unfocused, and unintelligible — lots of facts are listed, but no key point. This is unprofessional. Usually, an abstract consists of a background introduction, the current challenge, the key summary of the work, some highlighted results, and the potential impact of the work. Please refer to any well-written paper to find out what I mean.
- “Spectral curves characterizing the evolution of the calculated spectrum of the absorption cross-section depending on the variable length h of the GNS tips are shown in Figure 3b.” Please provide detailed information regarding your absorption cross-section calculation methods. Also, the simulation or calculation method of Figure 9 needs to provide.
- “In the present work, we evaluate the prospects of a new method for stimulating field and photoemission in planar-end blade emitters [1] using plasmonic nanostructures, which also provide optical control of tunneling emission by selecting femtosecond pulse irradiation modes. The motivation for conducting these studies is associated, on the one hand, with the trend of rapid dissemination and practical application of precision diagnostic systems using plasmonic nanoparticles (for example, SERS [11], visualization of biological tissue heterogeneities [12, 13] ……media (including catalysis [18, 19], ionization [20, 21], field emission of electrons [22−24])”. This sentence is extremely long and very difficult for the readers to understand. Please cut it into several short sentences.
- “It is expected that the use of plasmonic resonance nanoparticles at the “Mo-DLC- GNS” interface, due to the plasmon focusing properties of effective local absorption of optical radiation by GNS’s tips and, accordingly, the concentration of hot electrons in a strong electrostatic field, can become effective sources of electron emission, for example, for matrix devices generation of local X-ray radiation and ultrafast photodetectors”. The sharp tips are essential for the enhancement of tunneling current in this work due the giant field enhancement on the tips. Some recent works with strong field localization at plasmonic sharp tips may help the authors to support their claim. [Nano Lett.2020, 20, 6, 4550–4557; Nano Lett. 2020, 20, 9, 6351–6356].
- “The physical and technological aspects of the use of GNS for activating the tunnel emission of an electron current source based on planar-end blade structures [1] with a developed surface for possible applications in the field of biomedical instrumentation and sensing, including a matrix source with photoemission for an X- ray computed tomography sensors.” This sentence is ungrammatical.
Round 2
Reviewer 1 Report
The article has been systematically modified and can be accepted.
Reviewer 3 Report
The authors have addressed my comments and improved the manuscript substantially. The manuscript is scientifically sound. Thus, I have no further questions but to give my proposal of acceptance.